# *Cyprinus carpio* Skeleton Byproduct as a Source of Collagen for Gelatin Preparation

**DOI:** 10.3390/ijms23063164

**Published:** 2022-03-15

**Authors:** Robert Gál, Pavel Mokrejš, Jana Pavlačková, Dagmar Janáčová

**Affiliations:** 1Department of Food Technology, Faculty of Technology, Tomas Bata University in Zlin, Vavrečkova 275, 760 01 Zlin, Czech Republic; gal@utb.cz; 2Department of Polymer Engineering, Faculty of Technology, Tomas Bata University in Zlin, Vavrečkova 275, 760 01 Zlin, Czech Republic; 3Department of Lipids, Detergents and Cosmetics Technology, Faculty of Technology, Tomas Bata University in Zlin, Vavrečkova 275, 760 01 Zlin, Czech Republic; pavlackova@utb.cz; 4Department of Processing Control and Applied Computer Science, Faculty of Applied Informatics, Tomas Bata University in Zlin, Nad Stráněmi 4511, 760 05 Zlin, Czech Republic; janacova@utb.cz

**Keywords:** biopolymers, biotechnology, by-product, circular economy, collagen, *Cyprinus carpio*, fish, gelatin, skeletons, sustainable polymers

## Abstract

Byproducts obtained from fish processing account for up to 70% of their live weight and represent a large amount of unused raw materials rich in proteins, fats, minerals, and vitamins. Recently, the management of the use of predominantly cold-water fish byproducts has become a priority for many processing companies. This paper describes the biotechnological processing of byproducts of warm-water *Cyprinus carpio* skeletons into gelatins. A Taguchi experimental design with two process factors (HCl concentration during demineralization of the starting material and the amount of enzyme during enzyme conditioning of the collagen) examined at three levels (0.5, 1.0 and 2.0 wt%; 0.0, 0.1 and 0.2 wt% respectively) was used to optimize the processing of fish tissue into gelatin. Depending on the preparation conditions, four gelatin fractions were prepared by multi-stage extraction from the starting material with a total yield of 18.7–55.7%. Extensive characterization of the gel-forming and surface properties of the prepared gelatins was performed. Gelatins belong to the group of zero–low-medium Bloom value (0–170 Bloom) and low–medium viscosity (1.1–4.9 mPa·s) gelatins and are suitable for some food, pharmaceutical, and cosmetic applications. During processing, the pigment can be isolated; the remaining solid product can then be used in agriculture, and H_3_PO_4_Ca can be precipitated from the liquid byproduct after demineralization. The carp byproduct processing technology is environmentally friendly and meets the requirements of zero-waste technology.

## 1. Introduction

A byproduct is any other product that falls off during the production of the main product; it may or may not have a market value [1]. Depending on the state, we distinguish between solid, liquid, and mixed byproducts. The food industry produces byproducts in the processing of meat and fish, sugar, milk, and fruits and vegetables [2,3]. There are no hazardous waste materials produced during the processing of animal products into food. Certain byproducts can be used as livestock feed. On the other hand, wastewater poses a major burden on the environment and needs to be treated [4]. Leaving aside the risk groups of animal byproducts such as carcasses and parts of dead animals, animals infected with transmissible spongiform encephalopathies (TSE), tissues containing excessive amounts of contaminants, digestive tract contents, or byproducts collected during wastewater treatment, the bodies and parts of many slaughtered animals can be used for human consumption. This is covered by regulations such as Regulation (EC) No. 1069/2009 of the European Parliament and of the Council [5]. In practice, this is often not the case for cultural, customary, commercial or religious reasons. An exception is, for example, blood or intestines, which are used in many countries in the food and pharmaceutical industries.

Fish play an important role in providing food and nutrition worldwide, and are considered a key element of a healthy diet. Close attention is paid to fish as a source of essential nutrients in our diet thanks to their content of high-value proteins, a source of long-chain ω-3 fatty acids and microelements [6]. From 2017 to 2019, the average annual consumption of fish meat per person was 20.4 kg; in the European Union, it was about 11 kg, and in the USA about 8 kg, including seafood [7]. The world’s annual fish catch in recent years has exceeded 100 million tones, with about 80% marine fish and 20% freshwater fish. The main reason for the consumer preference for marine fish is the higher price of freshwater fish, which is caused by a longer breeding period (in ponds it can be up to four years). During this time, the fish must be fed and the ponds fertilized organically, and in addition fishermen must take care of the optimal environmental conditions in which the fish live. The considerable development, diversity, and popularity of marine fish with consumers points to the greater use of marine fish today. Greater use of freshwater fish is a great prospect for the processing and subsequent consumption of such a valuable source of nutrients.

Byproducts of fish processing are divided into liquid and solid states. Wastewater consists of pump water when unloading fish, rinsing water used for fish cutting (filleting, skinning, slaughtering, and other operations), and salt water (from salting and drying). The chemical composition of wastewater depends on the area of harvesting, the season, the species of fish, and the type and extent of treatment, and consists mainly of lipids, proteins, and minerals. Several methods have been proposed to obtain lipids from wastewater; hydrostatic pressure extraction to obtain ω-3 fatty acids appears to be the most effective [8]. Solid fish byproducts include skeletons, heads, fins, skins, and scales [9]. Many fish processing plants do not deal with recycling technologies, and wastewater is discharged into the aquatic environment while solids end up in landfills and veterinary remediation facilities; in both cases, it is a burden on the ecosystem and a waste of raw materials that contain valuable nutrients [10]. Fish byproducts, both organic and inorganic, have a high application potential. For example, waste myofibrillar proteins can be processed into transparent and flexible bioplastics by a suitable physico-chemical method with the addition of plasticizers, which can be used, for example, in packaging applications [11]. Fish oil rich is in polyunsaturated ω-3 and ω-6 fatty acids and can be used to make edible oils, fats, margarines, and food supplements (e.g., in the form of soft gelatin capsules), or used in cosmetic creams [12]. Methyl esters (biodiesel) can be prepared by acid- and alkali-catalyzed transesterification of waste fats. Waste fats can be further processed into soaps and balsams or used in the production of paints or varnishes. Inorganic fish byproducts are important as well. The cytoplasm contains chromatophores, which are pigment cells. The mutual layering of different types of chromatographs (silver, yellow, red and black pigments) causes the resulting structural coloration of the fish. These pigments can be used for the production of paints or coatings [13].

After technological processing of fish via scaling, disemboweling, and filleting (which is the primary technological section), up to 70% of solid byproducts remain. Collagen contained in various amounts in skeletons, heads, fins, skins, scales, or intestines can, after purification, be processed into gelatins and hydrolysates [14,15,16]. Gelatins are important biopolymers and hydrocolloids produced by the partial hydrolysis of collagen [17]. Although gelatins have a similar amino acid composition to collagens, they have different gel, surface, and rheological properties. The quality of gelatins is determined mainly by the gel strength, viscosity, melting and gelling points, water-binding ability, and foaming and emulsifying properties [18]. The unique thermoreversible properties of gelatin are used in many food, pharmaceutical, and medical applications [19]. The melting and gelling points of gelatins are below human body temperature in a relatively narrow temperature range (5–15 °C), which is more advantageous in many applications than other polysaccharide-based hydrocolloids [20,21]. The surface properties of gelatins are based on the presence of hydrophilic and hydrophobic moieties in their molecule, which tend to migrate to surfaces, thereby reducing the surface tension of aqueous systems and forming the desired identically charged film around the components of the dispersed phase. Gelatins with a higher isoelectric point value (pI > 7.0) are suitable for the formation of oil-in-water emulsions with a positive charge in a wider range of pH values, and thus exceed the common types of protein emulsifiers (soy, casein, or whey proteins) [22]. Film-forming applications of gelatins are used for the production of edible materials (films, foils, and coatings) for food and for the production of degradable matrices for biomedical applications [23].

The safety of gelatins is characterized mainly by microbiological limits (total aerobic microbial count, *Salmonella*, *Escherichia coli*, total yeast and molds count) and the highest permissible concentrations of heavy metals, mineral content, or pH value [24]. Commercial gelatins are made from mammal, (bovine and porcine) tissues [25,26]. Such gelatins are rejected by some consumers for dietary, religious, or cultural reasons. Fish or poultry gelatins can be a suitable alternative [27,28,29]. Compared to standard porcine and bovine gelatins, most fish gelatins have a lower gel strength, viscosity, and melting and gelling point, which limits their industrial use. In addition, gelatins extracted from fish living in warm water have better properties (especially gel-forming) than gelatins extracted from cold-water fish [30,31].

The preparation of fish gelatins is in principle the same as the preparation of gelatins from mammals and poultry. The starting tissue (or organ) rich in collagen must first be freed of accompanying non-collagenous proteins and fats using water, weak solutions of salts, acids, bases, and solvents. Purified collagen is subjected to controlled conditioning, mostly in an acidic environment, thus achieving partial chemical denaturation of collagen at the level of the quaternary structure. Consequently, during heat denaturation with hot water in a relatively narrow temperature range the non-covalent bonds of the tertiary structure of collagen are disrupted, resulting in the solubilization of the collagen and the preparation of the gelatin solution. In the case of fish collagen, which is characterized by a lower degree of intermolecular crosslinking, the pH is not as low or high as in the case of bovine or porcine collagen; in addition, the conditioning time is significantly shortened. The hot water extraction of gelatin takes place at lower temperatures (from 40 °C, usually up to 50 °C) than is usual for the extraction of gelatins from bovine and porcine tissues [32,33].

Although the current literature has more often mentioned the processing of fish byproducts for the preparation of collagen and its products (gelatins and hydrolysates) in the last decade, most authors focus on the processing of marine fish tissues such as cod, tuna, kerapu, kerisi, kembung, megrim, seabass, unicorn leatherjacket, and bigeye snapper [30,31,32,34,35,36,37]. Although fresh-water fish are a popular dish, especially for inland states, studies on the treatment of waste from their processing are, with few exceptions, rare, including the processing of carp byproducts [33,38,39,40,41]. Therefore, the main goal of our work was to prepare gelatins from carp (*Cyprinus carpio*) skeletons and to monitor the influence of selected process parameters in the processing of the starting material on the yield of gelatins. To ensure maximum conversion of the starting collagen material to gelatin, four gelatin fractions were extracted; sequential extraction is common in the industrial production of gelatins [26]. When designing the processing technology, we based on our experience in the preparation of gelatins from poultry byproducts using the biotechnological process of raw material conditioning, which is an environmentally friendly method [42,43]. With regard to the proposals of potential applications of prepared gelatins, especially in the food industry, another goal was to determine selected gel-forming properties of gelatins (gel strength, melting and gelling point, water-holding capacity, and fat-binding capacity), surface properties (foaming capacity and stability, emulsifying capacity and stability), and viscosity, which is especially important when choosing a suitable processing technology. Finally, the aim was to design optimal processing conditions for *Cyprinus carpio* skeletons in order to meet the parameters of zero-waste technology as much as possible. Scientific hypotheses: The proposed processing technology is expected to produce gelatin fractions from carp skeletons in similar yields as in the processing of other fish byproducts. It is assumed that the prepared gelatins will have similar properties as gelatins from traditional raw material sources or alternative raw materials (fish, poultry).

## 2. Results

The results of processing carp skeletons into four gelatin fractions are presented in the following four subsections.

### 2.1. Mass Balance of the Process

The schedule of experiments and results of the processing of *Cyprinus carpio* skeletons into four gelatins fractions are presented in Table 1. Table 2 shows the results of analysis of variance for four gelatin yields.

From the results of the regression analysis of gelatin yields, it is clear that both monitored process factors (concentration of HCl at collagen demineralization and the amount of enzyme during enzyme conditioning of collagen) are statistically significant (*p*-values < 0.05) in all four gelatin yields. The contour plot in Figure 1 shows the relationship between a response variable (gelatin yields) and two predictor variables (Factors A and B). From Figure 1a, it can be seen that the yield of the first gelatin fraction at the lower limits of both monitored factors (i.e., 0.50 wt% HCl concentration during demineralization of the raw material and zero amount of enzyme during enzyme conditioning of collagen) is low (<5.0%). With increasing HCl concentration and with increasing enzyme addition the yield increases significantly, and at 0.2 wt% of enzyme and HCl concentration > 1.4 wt% the yield of the first gelatin fraction is more than three-fold (>17.5%) greater. This is analogous to the yield of the second gelatin fraction, where at the lower limits of both predictor variables the yield is <5.0% and with increasing HCl concentration and with increasing enzyme addition it increases up to six-fold; see Figure 1b. At the upper limits of the monitored factors the yield of the second gelatin fraction equals approximately 30%. The trend is the opposite for the yield of the third gelatin fraction than for the previous two yields, and the maximum yield is at the lower limits of both monitored factors; see Figure 1c. Furthermore, it can be seen that the yield of gelatins from the third extraction step is very low (approximately 3% to 6%). We explain this by the high yields of gelatins from the first two extraction steps. The yields of gelatins obtained from the fourth extraction are similar to those from the third extraction, that is, up to about 7%; see Figure 1d.

If we compare the yields of gelatins prepared in individual extraction steps according to the Taguchi design (Experiments 1–9) with the yields of gelatins prepared according to the conditions of a blind experiment (Experiment 10), we find that both processing factors (Factors A and B) within the studied limits have a significant effect on the yield of gelatins. In the blind experiment, the yields of the four gelatin fractions (3.62%, 3.55%, 1.94%, and 1.86%, respectively) are always lower than for any gelatin prepared under experimental conditions 1–9. The total yield of all four gelatin fractions from the blind experiment is about 11.0%, while the total gelatins extraction yield of the experiments according to the studied design is 18.7–55.7% (see Table 1).

During the multistage extraction of gelatins, 0.1–1.0% of the green pigment and 0.2–2.0% of the residual fat (both based on purified collagen) were isolated from the purified collagen, depending on the process conditions (see Table 1). In addition, the separated liquid product (collagen hydrolysate) formed after biotechnological treatment of collagen represented, after drying, 0.7–17.8% by weight of the original purified collagen. The accuracy of the results of the mass balance of the process is confirmed by the mass balance error, the value of which was <2.5% in all ten experiments.

### 2.2. Gel-Forming Properties of Gelatins

The results of analyses of the gel-forming properties of four gelatin fractions prepared from *Cyprinus carpio* skeletons are shown in Table 3.

#### 2.2.1. Gel Strength

Table 4 shows the results of analysis of variance for four gelatin gel strengths.

The relationship between gelatin gel strengths and the studied processing factors (A and B) is presented by discrete contours in Figure 2. Only three gel-forming gelatins were prepared from the first gelatin fractions (Experiments 1, 2 and 4; see Table 3). From Figure 2a, it is clear that the highest quality gels (100–120 Bloom) are achieved at low HCl concentrations and minimal enzyme addition. From the *p*-values (see Table 4) it is clear that none of the process factors are statistically significant (*p*-values > 0.05). From the analysis of the gel strength values of the second gelatin fractions (see Figure 2b), it is clear that the gel strength values are relatively low. Both monitored process factors are statistically significant (*p*-values equal to 0.037 and 0.020). As with the first gelatin fractions, the gel strength increases with decreasing enzyme addition; a contour bounded by 0.00–0.10 wt% by the addition of enzyme and 1.05–2.00 wt% HCl represents the ideal process conditions to achieve a gel with a strength slightly exceeding 60 Bloom. The results of the regression analysis of the gel strengths of the third gelatin fractions show that both Factor A and Factor B are statistically significant (*p*-values 0.020 and 0.05). The trend of increasing gel strength is similar to the trend found in the second gelatin fraction, and the negative effect of higher enzyme addition on gel strength is evident. Figure 2c shows a gradual increase in gel strength (from values < 20 Bloom) while decreasing the amount of enzyme added and with increasing HCl concentration. Very decent values of gel strength (100–120 Bloom) are achieved with minimal enzyme addition (<0.01 wt%), while the HCl concentration does not affect the resulting strength. The situation is somewhat different for gelatin gels of the fourth fraction; see Figure 2d. From the positions of the contours, which are almost horizontally oriented, it is clear that the HCl concentration has essentially no effect on the strength of the gelatin gels; this fact is confirmed by its *p*-value of 0.770, which significantly exceeds the limit of statistical significance (0.05). In contrast, the addition of enzyme is statistically significant (*p*-value = 0.09). The highest gel strengths (around 150 Bloom) are achieved with very low enzyme addition (<0.01 wt%) and HCl concentrations between 0.8 and 1.8 wt%.

#### 2.2.2. Melting Point and Gelling Point

Melting point (M_P_) a gelling point (G_P_) could only be measured on gelatins that had a gelling ability. The relationship between M_P_ and the strength of gelatin gels is obvious in all gelatin fractions; as the strength of gelatin gel increases, M_P_ does as well. For the first gelatin fractions, M_P_ was measured only for Experiments 1, 2 and 4, and ranged from 22.5 to 29.4 °C. For the second gelatin fractions, M_P_ was measured for Experiments 1, 4, 5, 7 and 8. Due to the lower gel strength values for these gelatin fractions (10–65 Bloom), the M_P_ is also lower (18.6–24.8 °C) than for the first gelatin fractions. For the third gelatin fractions, M_P_ was measured in all experiments according to the Taguchi design except for the third, and was between 13.7 and 28.6 °C. For the fourth gelatin fractions, the gel strength was measured only for Experiments 1, 4 and 7, and M_P_ was relatively high, 26.7–29.9 °C. The G_P_ of the individual gelatin fractions differs as follows: first fraction 11.2–15.3 °C, second fraction 9.6–12.4 °C, third fraction 4.2–15.7 °C, fourth fraction 14.7–15.6 °C. The values of differences between M_P_ and G_P_ for individual gelatins from each gelatin fraction are interesting. This difference was 11.3–14.1 °C for gelatins from the first fraction, 9.0–12.4 °C for gelatins from the second fraction, 9.5–12.9 °C for gelatins from the third fraction and 12.0–14.3 °C for gelatins from the fourth fraction, which is about a 2–3-fold difference compared to bovine and porcine gelatin [26]. Gelatins prepared under blind experiment conditions (Experiment 10) did not form gels; therefore, M_P_ and G_P_ were not measured.

#### 2.2.3. Water Holding Capacity and Fat Binding Capacity

From the analysis of water holding capacity (WHC) values (see Table 3) the differences between gelatins from individual fractions are evident. In the first gelatin fractions, a very high WHC value was found for the gelatin prepared according to Experiment 1 (5.0 g/g); this gelatin is characterized by the highest gel strength (128 Bloom) of all first fraction gelatin. For other gelatins, with decreasing gel strength, WHC decreases, and for gelatins with zero gel strength WHC is 0.1–0.6 g/g. In contrast, for gelatins from the second gelatin fractions, WHC values are lower (0.2–2.0 g/g), and very low WHC values (0.2–0.3 g/g) for non-gel-forming gelatins were confirmed. The gelatins of the third fraction are comparable to those of the first fraction. High WHC (4.5 g/g) was found for the highest quality gelatin in this series (124 Bloom); as the gel strength decreases, the WHC decreases to 0.2 g/g. WHC gelatins prepared from the fourth fraction are the lowest, 0.3–1.0 g/g.

For fat binding capacity (FBC), the differences between gelatins from different fractions are not as significant. Once again, the trend was confirmed that gelatins with the highest gel strengths have the highest FBC. For the gelatin of the first fraction, the FBC ranges from low values (0.3 g/g) to an excellent value of 10.2 g/g, for the gelatins of the second fraction the FBC is 0.7–8.3 g/g, and for the gelatins of the third fraction the FBC values are even higher (0.9–11.1 g/g); gelatins of the fourth fraction are similar in this characteristic to those of the second fractions (0.7–7.4 g/g).

### 2.3. Surface Properties of Gelatins and Viscosity

The results of analyses of the surface properties of four gelatin fractions prepared from carp skeletons are shown in Table 5.

#### 2.3.1. Foaming Capacity and Stability

In the first gelatin fraction, foaming capacity (FC) was very low (3–8%); these gelatins have relatively low or zero gel strength (<45 Bloom). An exception was gelatin prepared under the conditions of Experiment 1 (lowest HCl addition and zero enzyme addition), which had FC = 20% (see Table 5); this gelatin had the highest gel strength (128 Bloom). A similar trend is evident in gelatins from the second fraction, as FC ranges from 4% to 20%; the highest value of FC was shown by gelatin with the highest gel strength (65 Bloom), while the lowest FC values were gelatin with zero gel strength (see Experiment 7 in Table 3). In addition, the gelatins of the first two fractions showed very low to zero foam stability (FS) values of 0–4%. For the gelatins of the third gelatin fractions, the lowest FC value was 3% (gelatin with zero gel strength) and the maximum FC value was 28% (124 Bloom gelatin). The highest achieved FC values were observed for gelatins from the fourth gelatin fractions, 52% for Experiment 4 (169 Bloom) and 48% in Experiment 7 (145 Bloom). As far as FS is concerned, the gelatins of the third fractions differed quite significantly, ranging from the minimum foam stability (1%) to a decent value of 20%. For gelatins of the fourth fraction, this difference was twofold: FS = 1–40%.

#### 2.3.2. Emulsifying Capacity and Stability

In contrast to foaming capacity, gelatins of all four fractions show a constant and very good emulsifying capacity (EC); for the first gelatin fractions 45.0–50.0%, for the second gelatin fractions 48.3–51.7%, for the third gelatin fractions 45.0–50.0%, and for the fourth gelatin fractions 48.3–55.0%. Emulsion stability (ES) achieves excellent values for gelatins across all fractions, from 96.6–100.0% for the first gelatin fraction, with a slight decrease for the second, third and fourth gelatin fractions (87.2–96.8%, 83.3–100%, and 90.9–100%, respectively; see Table 5).

#### 2.3.3. Viscosity

The viscosity values of gelatins correlate very well with the discovered values of gel strength. Gelatins from the first and second fractions belong to low-viscosity gelatins, in the first fraction it is 1.06 to 2.86 mPa·s; for gelatins of the second fraction, the viscosity is between 1.06 and 2.08 mPa·s. The situation is significantly different for gelatins of the third and fourth fractions. For gelatins of the third fraction, a very low viscosity value (1.07 mPa·s) was found for gelatin with zero gel strength (Experiment 3); as the gel strength increases, the viscosity does as well, and the highest viscosity value (4.50 mPa·s) was found for gelatin with 124 Bloom gel strength. For gelatins of the fourth fraction, approximately the same values were observed for gelatins prepared according to the conditions of Experiments 4 and 7 (4.88 and 4.59 mPa·s); these gelatins have a gel strength of 169 and 145 Bloom, while for gelatins with zero gel strength the viscosity is low (1.15–1.26 mPa·s).

### 2.4. Other Properties of Gelatins

Another important property that is part of gelatin product sheets is the ash content. The ash content of all four fractions was determined by the standard gravimetric method after burning and annealing the gelatin at 550 °C. Higher fractions of gelatins are always characterized by a lower maximum value of ash content. Gelatins from the first fraction had an ash content of 0.2–7.3%, gelatins from the second fraction 0.4–5.6%, gelatins from the third fraction 1.0–4.0%, and gelatins from the fourth fraction 0.6–3.0%.

## 3. Discussion

### 3.1. Mass Balance of the Process

Depending on the conditions of the process of preparation of gelatins from *Cyprinus carpio* skeletons, the four gelatin fractions were obtained with a total yield of 18.7–55.7%. Gelatin yields are significantly higher, e.g., compared to gelatin yields from skin of cuttlefish after alkaline conditioning of the raw material and extraction with various pepsin additions [44]; the authors report very low yields (2.7–9.2%). Very low yields of gelatins were achieved in the treatment of Nile tilapia skins (3.8%) and Nile perch skins (8.7%) after their treatment in weak solutions of NaOH and HCl [45]. Another study looked at the extraction of gelatin from tuna skin after different conditioning methods (CH_3_COOH concentration and treatment time) [29]; yields of 11.7–14.5% are relatively low, and lower than the lowest yield from our study (18.7%). Slightly higher yields were obtained with gelatins extracted from catfish skin; the authors focused on the study of the influence of different raw material conditioning methods and on different extraction parameters (temperatures of 60, 70 and 80 °C and times of 1, 2 and 3 h). When conditioned with 0.1 M NaOH, the yield of gelatins ranged between 7.0 and 19.5%, and when conditioned with 0.01% proteolytic enzyme solution (Alcalase) the yield increased to 10.5–23.3% [33]. The yield of about 20% was achieved in a study on the treatment of giant catfish skins after their conditioning in 0.005 mol/L CH_3_COOH and extraction at 40 °C [46]. In studying the effect of extraction temperature on gelatin yields from squid skin, the authors achieved very different gelatin yields; depending on the chosen extraction temperature (50–80 °C), it was 8.8–45.3% [47], and at higher extraction temperatures (60–80 °C) the authors achieved comparable yields of gelatins to those in our study. Similar results are reported by Arnesen and Gildberg in the extraction of gelatins from Atlantic salmon skins and Atlantic cod skins after two-bath tissue conditioning in 0.12M H_2_SO_4_ and 0.005M CH_3_COOH and two-stage extraction at 56 °C and 65 °C; gelatin yields were 39.7% and 44.8%, respectively [48]. Significantly different values of gelatin yields were observed during their sequential extraction at 50, 60, 70 and 95 °C from Nile perch skin; the total gelatin yield was 64.3% regardless of fish age [49], which is higher than in our studies. On the other hand, from the bones of the same fish yields were, 6.1–11.5% depending on the age of the fish, yields of gelatins which are about 3–6 times lower than the values achieved here. High yields were obtained in the extraction of gelatins from the skins of four species of marine fish (kerapu, jenahak, kembung, and kerisi) after two-bath conditioning in H_2_SO_4_ and citric acid [34]; yields of 43.6–68.5% are comparable to the total yield of gelatins in our Experiment 6 (44.2%), and about 1.2 times lower compared to the highest overall yield, which was achieved under the conditions of Experiment 9 (55.7%).

In conclusion, the treatment of carp skeletons using four-stage extraction leads to the preparation of four gelatin fractions with an overall yield of up to about 56%, which corresponds to the most observed yields in several studies of fish byproducts. The scientific hypothesis of gelatin yield has thus been confirmed.

### 3.2. Gel-Forming Properties of Gelatins

Tests of the gel-forming properties of gelatins were performed to confirm or refute the scientific hypothesis regarding the properties of gelatins prepared from *Cyprinus carpio* skeletons. The established hypothesis cannot be unambiguously confirmed or refuted, as many of the gel-forming properties of the gelatins were comparable to other fish gelatins or to gelatins prepared from traditional (bovine and porcine) raw materials, while other gel-forming properties were worse. A more detailed analysis is provided below.

#### 3.2.1. Gel Strength

According to gel strength, gelatins are usually divided into three categories: low-Bloom value gelatins (50–100 Bloom), medium-Bloom value gelatins (100–200 Bloom), and high-Bloom value gelatins (more than 200 Bloom) [26]. Depending on the conditions of the process of preparation of gelatins from *Cyprinus carpio* skeletons, four gelatin fractions were obtained with very different values of gel strengths, from immeasurable up to about 170 Bloom. Approximately half of all prepared gelatins did not form gels, about one third formed gels with a strength up to 100 Bloom, and about 20% had a gel strength above 100 Bloom.

From our study of the available literature on the processing of fish tissues into gelatins, it is clear that the quality of prepared gelatins, which is characterized mainly by gel strength, is influenced by many other factors as well, especially the species and age of fish, their living conditions (cold water, warm water, fresh water, marine), the type of tissue being processed (skin, bones), and further processing method, especially the method of conditioning (acids, bases, enzymes), extraction conditions (temperature, time, pH), and method of drying the obtained product. The gelatin yield obtained is a significant factor that further complicates the intercomparison; it is usually true that the high yield of prepared gelatin will have a negative effect on their quality parameters (gel strength, melting and gelling point, viscosity, etc.). From the results of many studies, it is clear that most of the prepared fish gelatins fall into the categories of low–medium Bloom value gelatin (gel strengths from about 70 to 200 Bloom); however, there are studies that mention the preparation of gelatins with gel strength values above 200 Bloom. Norziah et al. prepared gelatins with a gel strength of 69 Bloom from surimi processing waste (combined alkaline–enzyme conditioning and extraction at 50 °C) [30]. Low gel strength values (33–71 Bloom) were achieved when processing Atlantic cod skins into gelatins; on the contrary, higher gel strengths (80–108 Bloom) were achieved when Atlantic salmon skin was treated according to the same procedure [48]. Gelatin with a gel strength of 85–132 Bloom was prepared by processing squid skin at different extraction temperatures [47]. Most fish gelatins prepared according to our technology have comparable gel strength to gelatins prepared under the conditions of the above-mentioned studies. The previously-mentioned factors influencing gel strength are evidenced by a study describing the preparation of gelatins from the skins of four species of marine fish [34]. Depending on the type of skin being treated, the gel strength varies between 46 and 252 Bloom. Gelatins with good gel strength (153 Bloom) and relatively low yield (20.1%) were prepared by conditioning giant catfish skins in 0.005 mol/L CH_3_COOH and extracting at low temperature (40 °C) [46]. The relationship between lower gelatin yield and higher gel strength is confirmed by the findings of a study on processing cuttlefish into gelatin by alkaline conditioning and extraction with various proteolytic enzyme additions. The authors achieved good gel strength values (120–198 Bloom), although unfortunately with very low yields (2.7–9.2%) [44]. Almost identical gel strength values are reported by Pradarameswari et al. [33]. By processing the raw material with a proteolytic enzyme, 121–193 Bloom gelatin can be prepared; when treated with NaOH, the values are slightly higher (134–208 Bloom) with a yield of 7.0–19.5% and 10.5–23.3%, respectively. The complexity of processing fish tissues into gelatins is evidenced by the results of processing Nile perch skin and bones by sequential extraction at 50, 60, 70 and 95 °C [49]. For skin gelatins, a very wide range of gel strengths (81–208 Bloom) depends on both processing conditions and on the age of the fish; the yield is high (64.3%). For bone gelatins, it was 134–179 Bloom, though at very low yields (6.1–11.5%).

#### 3.2.2. Melting Point and Gelling Point

Compared to common commercial types of gelatins (porcine and bovine), where melting point (M_P_) values are usually between 21.0–34.0 °C [26], the M_P_ is lower for several gelatins prepared here, although in most cases it is comparable (13.7–29.4 °C). Compared to other studies on the preparation of fish gelatins, the M_P_ of gelatins prepared according to our technological procedure is comparable. For gelatins prepared from surimi processing waste, the M_P_ was only 16.2–18.9 °C [30], from cuttlefish 19.2–25.4 °C [44], from Nile perch skin 21.4–26.3 °C, from Nile perch bones 25.5–26.5 °C [49], from tuna skin 25.6 °C [29], from Nile perch skin 21.4 °C, and from Nile tilapia skin 28.5 °C [45].

As far as G_P_ is concerned, the situation is the opposite; all our prepared gelatins from any gelatin fraction have a lower G_P_ (4.2–15.7 °C), in many cases even more pronounced than with standard bovine and porcine gelatins, for which the G_P_ is usually about 5 °C lower than the M_P_. Compared to other fish gelatins, the gelatins we prepared have a comparable or slightly lower G_P_; gelatins prepared from cuttlefish had G_P_ 12.9–21.8 °C [44], from Nile perch skin 13.8–19.5 °C, from Nile perch bones 18.5–19.0 °C [49], from Nile perch bones 19.9 °C, from tuna skin 19.0 °C [29], and from Nile tilapia 17.1 °C [45]. Exceptions are gelatins prepared from surimi processing waste, for which the authors found a very low G_P_, 5.1–5.7 °C [30].

#### 3.2.3. Water Holding Capacity and Fat Binding Capacity

Gelatin fractions show different values of water holding capacity (WHC) depending on the preparation conditions, in the range of 0.1–5.0 g/g. Although WHC is an important parameter of gelatin, especially for some food applications, only a few studies are devoted to its determination. For Nile tilapia skins and Nile perch skins, the WHC values are 6.4 g/g and 6.9 g/g, respectively [45]; several of our gelatins approach these values. Gelatins prepared from yak skins showed very low WHC values; depending on the preparation and drying conditions it was 0.40–0.47 g/g, which are relatively low values compared to our gelatins [50]. In our previous study on the preparation and characterization of chicken skin gelatins, we found WHC 3.85–5.58 g/g, depending on extraction conditions [51], which are almost the same values as for our highest-quality fish gelatins. In a previous study, we tested high-quality porcine (type A 260 Bloom) and bovine (type B 260 Bloom) gelatins; values of 4.43 g/g and 6.42 g/g are comparable to our highest-quality fish gelatins. WHC 1.9 g/g in chicken collagen hydrolysates prepared by enzymatic hydrolysis from chicken feet [52] is lower than in our highest quality gelatins. Data from protein hydrolysates prepared from chicken eggs are known as well; a WHC between 1.9–2.9 g/g corresponds to some of our gelatins [53].

In terms of fat binding capacity (FBC), similar to WHC the values of the gelatins prepared here vary considerably depending on the preparation conditions, ranging between 0.3 and 11.1 g/g. The comparison with the literature data is as follows. For Nile tilapia skins and Nile perch skins, the FBC values are 4.7 g/g and 3.6 g/g [45], which are values comparable to our gelatins. Gelatins prepared from yak skins showed very low FBC values; depending on the preparation and drying conditions it was 0.16–0.20 g/g [50]. In our aforementioned study on chicken skin gelatins, FBC was very low (0.97–1.26 g/g) compared to fish gelatin, while commercial porcine (260 Bloom) and bovine gelatins (260 Bloom) did not differ significantly (0.42 g/g and 0.71 g/g) [51]. Dhakal et al. report an FBC of 5.3 g/g for collagen hydrolysates [52], and similar values are reported by Surangna and Anal for chicken egg hydrolysates (2.5–4.4 g/g) [53], which are parameters corresponding to our average quality gelatins. It is worth noting the low FBC values (0.21–0.29 g/g) for low molecular weight hydrolysates from yak bones [54].

### 3.3. Surface Properties of Gelatins and Viscosity

Gelatin surface tests were performed to confirm or refute the scientific hypotheses regarding the properties of gelatins prepared from *Cyprinus carpio* skeletons. As in the case of the gel-forming properties of gelatins the hypothesis cannot be unambiguously confirmed or refuted, as some of the surface properties of gelatins (emulsifying capacity and stability, viscosity) are comparable to other fish gelatins or gelatins prepared from traditional (bovine and porcine) raw materials and other surface properties (foaming capacity and stability) are worse. A more detailed analysis is provided below.

#### 3.3.1. Foaming Capacity and Stability

Depending on the preparation conditions gelatins with the required foaming capacity (FC) can be prepared, ranging from very low values (3%) to relatively good foaming values (52%). Regarding the stability of whipped foams (FS), there are fundamental differences across the gelatins of the four prepared fractions. Several gelatins prepared from the third and fourth fractions have good foam stability (20–40%); others, especially gelatins of the first and second fractions, have very low (<4%) or zero FS values.

Comparing the FC and FS results of our gelatins with the available data from fish gelatin preparation studies, it can be said that the foaming properties of our gelatins are rather average, which may not be a disadvantage, as high FC and FS values are unsuitable for certain gelatin applications. Giant catfish skin gelatin FC is 130%, and the FS 35% [46]. Excellent FC and FS values were obtained for gelatins extracted from squid skins (117–207% and 100–162% respectively) depending on the preparation conditions [47]. Similarly high FC and FS values are reported by Jridi et al. al for cuttlefish gelatins, 100–200% depending on the preparation conditions; however, it should be noted that gelatin yields were very low (2.7–9.2%) [44]. In contrast, very low values of FC (2.3–2.5%) and FS (1.9–2.0%) were observed for gelatins prepared from fresh-water fish skins of Nile tilapia and Nile perch, which is significantly less than our best gelatins [45]. Compared to other alternative and traditional types of gelatins, our fish gelatins are comparable. Depending on the preparation conditions, chicken skin gelatins have FC 35.6–61.1% and FS 4.4–38.9%; porcine gelatin (260 Bloom) has FC 62.2% and FS 14.4%, and beef gelatin (260 Bloom) has 55.1% and 13.2% [51].

#### 3.3.2. Emulsifying Capacity and Stability

The emulsifying properties of all prepared gelatins do not show significant differences and are characterized by very good emulsion capacity (EC) values of 45–55% and excellent emulsion stability (ES) values of 83–100%.

Information on EC and ES of fish gelatins is rare in the literature, although comparisons with other types of gelatins or hydrolysates are possible. The EC for hydrolysates prepared from capelin is 51% and the EC is 92% [55]. Li et al. report EC values of 57% and EC values of 59–72% (depending on preparation conditions) for low molecular weight hydrolysates from yak bones [54]. For chicken skin gelatins, depending on the preparation conditions the EC is 35.0–61.1% and FS is 72.5–87.5%; porcine gelatin (260 Bloom) has EC 30.7% and EC 94.4% and bovine gelatin (260 Bloom) has 57.7% and 88.9% [51]. It is clear that our fish gelatins are fully comparable in emulsification properties to other gelatins and hydrolysates.

#### 3.3.3. Viscosity

From the available literature, it is clear that the viscosity of fish gelatins ranges from low-viscosity gelatins (1.5–3.5 mPa·s) through medium-viscosity gelatins (3.5–5.5 mPa·s) to high-viscosity gelatins (5.5–7.5 mPa·s) [26]; e.g., gelatins prepared from black and red tilapia skin have a viscosity of 3.2–7.1 mPa·s [56]. Very good viscosity values (6.0 mPa·s) for fresh-water fish gelatins are reported by Ninan et al. [41]. The process parameters affect the viscosity of the gelatins prepared here, which vary across a wide range from 1.06 to 4.88 mPa·s. Prepared gelatins can be classified as low–medium viscosity gelatins, which are found in many food, pharmaceutical or medical applications [22,24].

### 3.4. Other Properties of Gelatins

Gelatin fractions show different values of ash content; depending on the preparation conditions, it is 0.2–7.3% based on gelatin dry matter. For the use of gelatins in food and pharmaceutical applications, gelatins must meet maximum ash limits, usually 3.0% and 2.0%, respectively, set by the relevant pharmacopoeias and food standards bodies [57,58]. For gelatins with above-limit ash content it is necessary to include deionization, which is a common process operation in the production of commercial gelatins [25,26]. When directly comparing the ash content of our gelatins with other fish gelatins, it is necessary to take into account the raw material source from which the gelatins are prepared. Fish skins, along with porcine, bovine, or poultry skins, are characterized by a low ash content. Thus, gelatins extracted from fish skins usually have a very low ash content, e.g., from cuttlefish skin (0.03–0.06%) [44] or from squid skin (0.2–0.7%) [47]. An exception is, for example, a study on the preparation of gelatins from Nile perch skin and bones, where the authors report a slightly higher ash content in skins (0.5–1.7%); however, the ash content in bones is high (4.4–11.2%) [49].

### 3.5. Discussion Summary

From the discussion of the mass balance results of the process of preparation of four gelatin fractions from *Cyprinus carpio* skeletons and from analyses of their gel-forming and surface properties in comparison with gelatins obtained from porcine, bovine, poultry, and other fish tissues, the considerable complexity of gelatin preparation from different collagen sources is evident, with consideration primarily of the nature of the extracted collagen tissue, type of collagen, and amino acid composition [59]. The structure of collagen, that is, the degree of inter- and intra-molecular crosslinking, is influenced mainly by the age of the animal, and thus has a fundamental influence on the physical, mechanical, and chemical properties of collagen [60]. Other factors influencing the chemical composition and structure of collagen include animal and fish farming conditions (e.g., nutrition, altitude, geographical location), and in fish there are differences between marine fish and fresh-water fish collagen. The structure and stability of collagen significantly influences the choice of process conditions for the preparation of gelatins based on chemical and heat denaturation of collagen quaternary and tertiary structures [61]. Different conditions of collagen conditioning in acidic or alkaline environments or using proteolytic enzymes and gelatin extraction (extraction method, temperature, time, pH) affect the molecular weight (mixture of polypeptide collagen chains) and amino acid composition of gelatin [62,63].

The chemical composition and structure of gelatins are among the many important factors influencing gel-forming capacity, viscosity, and surface properties of gelatins [64]. This is evidenced by the different functional properties between the groups of four gelatin fractions prepared at different extraction temperatures, as well as between the gelatins of one fraction. This is quite evident with gelatin gel strength. The gel strength is mainly influenced by the representation of chains with molecular weight (M_r_) approximately 100 kDa. Due to the fact that gelatins prepared here with measurable gel strength belong to the low–medium Bloom value gelatins, it is possible to assume a higher proportion of chains with M_r_ < 100 kDa. A higher proportion of chains with lower M_r_ result in weaker forming helix-like structures (clusters) during the cooling of the gelatin solution and the formation of weaker 3-D structures (junction zones) which subsequently arise from these clusters. In terms of amino acid composition, the content of imino acids (proline and hydroxyproline), which form nucleation sites through H-bridge formation, is important for these structural changes; they are therefore crucial amino acids for the creation of junction zones of emerging gel structures. Porcine and bovine gelatins are characterized by a higher content of imino acids [62,65,66]. The assumption of the influence of M_r_ and the content of imino acids is evident on other gel-forming properties of prepared gelatins, namely, gelling point (G_P_) and melting point (M_P_). Due to the lower M_r_ chains, such significant structural changes (formation of H-bridges and formation of hydrophobic interactions between gelatin molecules) involved in the stabilization of gel structures do not occur during cooling of the sol as in medium–high Bloom value porcine and bovine gelatins, in which stronger junction zones are formed and in higher numbers, which results in higher G_P_ values (approximately 35 °C) as well as higher values of other gel-forming properties of gelatins [67,68]. The G_P_ of our prepared gelatins is thus relatively low compared to porcine and bovine gelatins. On the contrary, in comparison with gelatins prepared from fish collagens, G_P_ is minimally comparable, which is due to the related amino acid composition, or the comparable content of amino acids proline and hydroxyproline [69,70]. The complexity of gelatin gel structures of prepared four gelatin fractions is clearly represented by the results of M_P_ of gelatins as well. As previously mentioned, while the difference between G_P_ and M_P_ is approximately 5 °C for commercial porcine or bovine gelatins [26], this difference was 2–3 times higher for our gelatins. Nevertheless, the dependence between gel strength versus G_P_ and M_P_ was confirmed in our prepared fish gelatins. Higher gel strength is connected to higher G_P_ and M_P_; the same trend is obvious in other types of gelatins [71]. Regarding the summary of surface properties of prepared gelatins, it is not possible to unambiguously interpret the differences in tested properties between groups of four gelatin fractions without information on the secondary structure of gelatins from which data on the percentage of individual components (triple helix, β-sheet, random coil) can be obtained [72]. This applies, for example, to selected gelatins of the third and fourth fractions, which show higher values of foaming capacity (FC) and significantly higher values of foaming stability (FS) than gelatins of the first and second fractions. We assume that this is related to the different amino acid composition (hydrophilic and hydrophobic amino acids) of gelatins, respectively, the nature of their side chains and the molecular structure of gelatins [73]. On the contrary, there are no major differences in the emulsifying properties (emulsion capacity and stability) between the groups of four gelatin fractions. The viscosity of gelatins is significantly affected by the presence of macromolecular chains with a high M_r_ (200–400 kDa), the presence of which depends on the method of conditioning the collagen raw material [74,75]. Differences in viscosity values are evident between the groups of four gelatin fractions. Similar to FC/FS, selected gelatins of the third and fourth fractions with a viscosity of up to 4.88 mPa.s belong to the category of medium-value viscosity gelatins, while the gelatins of the first and second fractions with a maximum viscosity of 2.86 mPa.s belong to category of low-value viscosity gelatins. However, the viscosity values correspond very well with the gelatin gel strengths; the highest values of gelatin viscosity always correspond to the highest values of gel strength (see results in Table 3 and Table 5).

### 3.6. The Importance of Work Results for Pratice

The processing of fish tissues from freshwater and marine fish into gelatin has been the subject of a number of studies. The vast majority of studies are devoted to the most accessible and most easily-processed tissue, skin [29,33,34,44,45,46,47,48,49,76]; few studies have addressed processing of heads [77,78] or bones [49]. These authors mostly use weak acid solutions (CH_3_COOH, H_2_SO_4_ or HCl) when conditioning the raw material (the process of breaking the quaternary structure of collagen); a rarity is the combined acid-alkaline method [45] or the use of a proteolytic enzyme [33].

The significance of our work lies in the fact that it deals with the processing of waste *Cyprinus carpio* skeletons into gelatins; this represents a non-traditional biotechnological method using a commonly available food proteolytic enzyme in the technological process of converting collagen into gelatin. In contrast to the above-mentioned studies, another new element is the treatment of conditioned collagen by four-stage extraction, which significantly increases the overall yield of the prepared gelatins. The four gelatin fractions differ in their surface and gel-forming properties; the properties are comparable with other fish gelatins as well as with gelatins from other alternative sources, e.g., poultry [79,80].

Considering the zero–low–medium gel strength and low–medium viscosity of our gelatins, we propose their use in the food industry for the production of certain types of confectionaries (e.g., meringues, chewy candies, caramels, and toffees) and dairy products (creams for long storage, butter-type spreads, puddings, and acidified milk products). Thanks to the very good emulsifying capacity and emulsion stability, certain gelatins can be used in the production of meat products (e.g., pâtés). Due to the lower melting and gelling point values of our gelatins, they cannot be used for the preparation of jellies or gummy bears; on the contrary, they are very good as an additive in the production of ice cream, or as a wine and fruit clarification agent. It is worth mentioning the possible use as a packaging material in the production of micro-capsules to encapsulate various food additives [19,22]. Our gelatins can be used in pharmaceutical applications as a bonding agent in the manufacture of tablets and subsequently for coating tablets to reduce dust, mask unpleasant tastes, and allow printing and color coating for product identification. In medicine, it can be used for the production of hydrogels and nanofibers, as a matrix for intravenous infusions, as injectable microcryogels, and in the production of surgical dressings [24]. It is worth mentioning that fish gelatins are beneficial for cosmetic applications, for which zero–low bloom value and low viscosity value gelatins or gelatin hydrolysates are suitable; they may be used as additives to skin creams to improve the hydrating properties of the *stratum corneum* or to reduce trans-epidermal water loss. Medium Bloom value gelatin can be used as a gelling agent in bath salts, shampoos, sunscreens, body lotions, hair sprays, etc. [81].

During the processing of *Cyprinus carpio* skeletons into gelatins a number of byproducts are formed which can subsequently be used. Maceration liquor, a byproduct formed during the separation of inorganic components from the raw material, can be processed by precipitation with Ca(OH)_2_ to CaHPO_4_.2H_2_O, which can be used as an additive in feed mixtures or in the production of plant growth stimulators. Collagen hydrolysate, a byproduct formed after biotechnological treatment of collagen, can be used in the food industry as a food supplement, in cosmetic matrices, as an ingredient in livestock feed, and in the production of fertilizers. Pigment, a by-product formed after centrifugation of gelatin fractions, can be used in the production of paints and coatings. When centrifuging the gelatin fractions, the residual fat is separated, which can be used as a nutritional supplement; it contains ω-3 fatty acids. The remaining undissolved residue, a byproduct formed after extraction of the last gelatin fraction, could be processed by hydrolysis into collagen hydrolysate, or used directly in agriculture as a nitrogen-rich fertilizer.

In further research into the processing of *Cyprinus carpio* skeletons into gelatins, if desired for the final applications of gelatins we recommend focusing on the improvement of specific gel-forming properties of gelatins, especially melting point and gelling point, or alternatively gel strength as well as surface properties, especially foaming capacity and stability, e.g., by adjusting the conditions in the conditioning of the raw material, in the extraction of gelatins, and in the finalization of gelatins. Yang et al., who prepare gelatins from Yak skin, have shown that a suitable drying method can significantly improve the functional properties of gelatins; they favor freeze drying as the best method [50].

## 4. Materials and Methods

### 4.1. Materials, Appliances and Chemicals

Carp skeletons were obtained from Tovačov Fisheries (Tovačov, Czech Republic). Skeletons are a solid byproduct of the preparation of carp fillets which, in addition to the bones, contains residues of muscle, skin, scales, and fins. The skeletons were cut into smaller pieces and further ground on a meat grinder about 3 mm in size. The ground raw material was deaerated (−0.6 bar) on a vacuum device by sealing in a vacuum bag in amounts of approximately 1 kg, and the packages were frozen at −18 °C. Prior to processing, the frozen raw material was thawed in a refrigerator at 5.0 ± 1.0 °C for 24 h. First, byproduct material analyses were performed by conventional food methods [82,83,84]. Dry matter content was 43.0 ± 0.5%; furthermore, within the dry matter fibrous protein was 20.6 ± 0.4%, the collagen proportion in the fibrous protein was 73.4 ± 1.1%, fat 59.8 ± 0.5%, inorganic solids 9.4 ± 0.2%, and globular proteins and glycoproteins 10.4 ± 1.8%. Each analysis was repeated three times, and mean values and standard deviations were calculated.

We used a Stevens LFRA Texture Analyzer for measuring gelatin gel strength (Leonard Farnell and Co Ltd., Liverpool, UK) as well as a Ubbelohde viscometer (Technisklo Ltd., Držkov, Czech Republic), Braher P22/82 meat mincer (San Sebastian, Spain), Nedform LT 43 shaker (Nedform Ltd., Valašské Meziřící, Czech Republic), Kern 440–47 electronic scale, Kern 770 electronic analytical balance (Kern Ltd., Balingen, Germany), analytical mill IKA A 10 labortechnik (IKA Werke Ltd., Staufen, Germany), Memmert ULP 400 drying oven (Memmert Ltd., Bűchenbach, Germany), Samsung fridge freezer (Samsung Electronics Co. Ltd., Seoul, Korea), Henkelman Boxer 42 vacuum packaging machine (Henkelman Vacuum Systems, CK ‘s-Hertogenbosch, The Netherlands), Rotina 35 centrifuge (Hettich Ltd., Berlin, Germany), IKA T 25 digital Ultra-Turrax (IKA-Werke Ltd., Staufen im Breisgau, Germany), Whatman no. 1 paper (Sigma Aldrich, Gillingham, UK), WTW pH meter Multical pH 526 (WTW Ltd., Weilheim, Germany), heating board Schott Geräte (Schott Geräte Ltd., Mainz, Germany), a metal filter sieve with pore size 1 and 2 mm (Labor-komplet, Praha, Czech Republic), and laboratory glassware.

Chemicals used were NaCl, NaOH, HCl, petroleum ether, and ethanol (Verkon, Praha, Czech Republic); all chemicals were of analytical grade. Protamex^®^, Novozymes endoprotease (Copenhagen, Denmark) was used for conditioning of purified collagen from *Cyprinus carpio* skeletons; it is a *Bacillus* protease complex with declared activity 1.5 AU/g. Optimal working conditions are pH 5.5–7.5 and temperature < 60 °C. The enzyme complies with the recommended purity specifications for food-grade enzymes issued by the Joint FAO/WHO Expert Committee on Food Additives (JECFA) and the Food Chemicals Codex (FCC).

### 4.2. Design of Eperiments and Statistical Analysis

Design of experiments (DOE) makes it possible to find the process factors that most significantly affect the production process and its outputs and to determine the optimal parameters of these factors. It is a mathematical tool that allows quantification of the significance of the inputs that are identified at the beginning of the experiment as probably being the most influential. Experimental planning enables the technological process to achieve the required outputs with minimal variability and resistance to unpredictable negative effects [85]. The Taguchi design of experiments was used in this study, in which two factors were monitored at three levels, 3^2^ (9 experiments). Factor A was HCl concentration during demineralization of the raw material: 0.5, 1.0, and 2.0 wt%. Factor B was the amount of enzyme during enzyme conditioning of collagen: 0.0, 0.1, and 0.2 wt%, based on dry weight of purified collagen. To contrast the results, a blind experiment was performed (Experiment 10); during demineralization of raw material, pure H_2_O was used instead of HCl, and during enzyme conditioning of the collagen, no enzyme was added.

Analyses of gelatins were performed in triplicate; mean values were calculated using Microsoft Office Excel 2013 (Microsoft, Denver, CO, USA). To evaluate the obtained data, regression analysis was applied to all results using Minitab^®^ 17.2.1 statistical software for Windows (Fujitsu Ltd., Tokyo, Japan). The level of significance was set to 5%, *p*-value ≤ 0.05; factors with a value ≤0.05 have an effect on the evaluated process variables with 95% probability. Contour plots (Figure 1 and Figure 2) showing the relationship between a response variable and two predictor variables (Factors A and B) by viewing discrete contours of the predicted response variable were evaluated using the same software.

### 4.3. Processing of Cyprinus carpio Skeletons into Gelatins

The scheme of complex processing of carp skeletons into four fractions of gelatins, including other usable products created during processing, is shown in a flow chart in four technological sections; see Figure 1.

In the technological process of processing *Cyprinus carpio* skeletons into four fractions of gelatins, it is first necessary to separate a number of accompanying components of organic and inorganic origin from the starting raw material. The purified collagen is then biotechnologically treated with food protease (disruption of the quaternary structure of collagen) and converted into liquid forms (gelatin solutions) by multistage hot water extraction (disruption of the tertiary structure of collagen); see details below.

Separation of organic matter. The thawed raw material was first washed with cold H_2_O. It was mixed with 0.2 mol/L NaCl in a ratio of 1:6 and shaken at room temperature (23.0 ± 1.0 °C) for 90 min, then washed with cold H_2_O. It was then mixed with 0.03 mol/L NaOH in a ratio of 1:6 and shaken at room temperature for 45 min and, after filtration, washed with cold H_2_O; this procedure was repeated three more times. Finally, the raw material was washed with cold H_2_O and dried at 35 °C for 24 h. This was followed by a defatting step; the raw material was mixed in a ratio of 1:9 (*w*/*v*) with petroleum ether and ethanol (mixed in a ratio of 1:1, *v*/*v*) and shaken for 48 h at room temperature, then after 12 h the solvent was changed.Separation of inorganic matter. The raw material was mixed in a ratio of 1:10 with HCl (concentration according to Factor A) and demineralized with gentle shaking at room temperature for 48 h; after 24 h, the acid was replaced with new acid. After filtration, the demineralized collagen was washed thoroughly with cold H_2_O and dried for 24 h at 35 °C.The purified collagen was mixed with H_2_O in a ratio of 1:15 and after shaking for 20 min, the pH was adjusted to 6.5–7.0 (by adding 5 wt% NaOH solution). The proteolytic enzyme was then added in an amount according to Factor B and the mixture was shaken at room temperature for 4 h; at 30-min intervals, the pH was checked (and adjusted) to the prescribed range. After filtering off the liquid product (collagen hydrolysate), the collagen was thoroughly washed with cold H_2_O. Collagen hydrolysate was dried in a thin layer (4 mm) in a circulating air drier at 50.0 ± 0.5 °C for 20 h.Multistage extraction of gelatins. Biotechnologically treated collagen was subjected to four separate sequential extraction cycles using a batch process extractor. In the first extraction stage, collagen was mixed with H_2_O in a ratio of 1:15 and the mixture was heated while stirring at a rate of dt/dτ = 10 °C/min to a temperature of 40.0 ± 0.5 °C, at which point the extraction lasted 45 min. After filtration, the solution of the first gelatin fraction was immediately heated to a temperature of 95.0 ± 0.5 °C (dt/dτ = 15 °C/min) and maintained at this temperature for 5 min; the residual enzyme was inactivated in this way. After cooling to room temperature, the gelatin solution was centrifuged at 4000 rpm for 4 min; the pigment (bottom layer) and residual fat (top layer) were separated from the gelatin solution. The gelatin solution was poured into a thin film (4 mm) and cooled into a gel in a refrigerator at 5.0 ± 0.5 °C for 30 min and then dried in a circulating air drier, first at 40.0 ± 0.5 °C for 12 h, then at 65.0 ± 0.5 °C for 8 h. The resulting gelatin film was scraped off, weighed and ground to a powder. In the second, third, and fourth extraction stages, the same procedure was followed at extraction temperatures of 50.0 ± 0.5 °C, 70.0 ± 0.5 °C and 95.0 ± 0.5 °C. The second and third gelatin fractions were inactivated in the same way as the first gelatin fraction. After centrifugation, less pigment and residual fat were separated from each subsequent extraction. The undissolved residue remaining after the fourth extraction cycle was dried at 103.0 ± 1.0 °C to a constant weight and then weighed; the pigment was dried overnight at 40.0 ± 1.0 °C and then weighed. The prepared gelatins were then subjected to further analyses.

### 4.4. Analytical Part

The yield of the hydrolysate (Y_H_) was calculated from the weight of the hydrolysate prepared after conditioning of the raw material based on the initial weight of the purified collagen (Equation (1)) and the yield of gelatins (Y_G1_, Y_G2_, Y_G3_, Y_G4_) from the weight of extracted gelatins based on the initial weight of the purified collagen (Equation (2)). According to the same principle, the yields of pigment and residual fat were calculated (Equations (3) and (4)). Further, the total gelatins extraction yield (∑Y_G_) and the portion of undissolved residue (UR) were calculated (Equations (5) and (6)). The mass balance error (MBE) is expressed by the % difference of the dry matter mass balance between the input (purified collagen) and the outputs (hydrolysate + gelatins + pigment + fat + undissolved residue); see Equation (7).
Y_H_ = (m_H_/m_0_)·100,(1)
Y_G_ = (m_G_/m_0_)·100,(2)
Y_P_ = (m_P_/m_0_)·100,(3)
Y_RF_ = (m_RF_/m_0_)·100,(4)
ΣY_G_ = Y_G1_ + Y_G2_ + Y_G3_ + Y_G4_,(5)
UR = (m_UR_/m_0_)·100,(6)
MBE = [100 − (Y_H_ + Y_G1_ + Y_G2_ + Y_G3_ + Y_G4_ + Y_P_ + Y_RF_ + UR)],(7)
where Y_H_ is the yield of hydrolysate (%), Y_G1_ is the yield of the first gelatin fraction (%), Y_G2_ is the yield of the second gelatin fraction (%), Y_G3_ is the yield of the third gelatin fraction (%), Y_G4_ is the yield of the fourth gelatin fraction (%), Y_P_ is the yield of pigment (%), Y_RF_ is the yield of residual fat (%), UR is an undissolved residue (%), m_0_ is the weight of the purified collagen (g), m_H_ is the weight of the hydrolysate (g), m_G_ is the weight of gelatins (g), m_P_ is the weight of pigment (g), m_RF_ is the weight of residual fat (g), and m_UR_ is the weight of undissolved residue (g).

Gel strength, viscosity, and ash content were determined according to the standard testing methods for edible gelatins [86]. Because these are common gelatin test methods, we only present their principle. The gelatin gel strength (GGS) was determined from a gel formed from a 6.67% solution prepared according to prescribed conditions by measuring of the force (weight in grams) required to depress a prescribed area of the surface of the sample to a distance of 4 mm. Viscosity (υ) of a 6.67% gelatin solution was determined by Ubbelohde viscometer. Ash content was determined gravimetrically after burning and annealing the sample. The following gelatin properties are not described in standard gelatin testing methods; thus, a brief test procedure is be provided here.

Gelatin water holding capacity (WHC) was determined according to Nasrin et al. [87] with slight modifications. In a plastic test tube, 1.0 g of the gelatin sample was mixed with 25.0 mL of distilled H_2_O and the contents were shaken vigorously for 5 min at room temperature. The contents of the test tube were centrifuged at 5000 rpm for 30 min, then the supernatant was filtered through Whatman no. 1. filter paper. WHC (g/g) was calculated from the weight of water absorbed by the gelatin sample, w_1_ (g), based on the weight of gelatin weighed, w_0_ (g); see Equation (8).
WHC = w_1_/w_0_,(8)

Gelatin fat binding capacity (FBC) was determined according to Li et al. [54]. In a plastic test tube, 0.1 g of the gelatin sample was mixed with 10.0 mL of sunflower oil and the contents were shaken vigorously for 30 min at room temperature. The contents of the test tube were then centrifuged at 2500 rpm for 30 min, and the supernatant was pipetted off and weighed. FBC (g/g) was calculated from the weight of oil absorbed by the gelatin sample, w_2_ (g), based on the weight of gelatin weighed, w_0_ (g), and multiplied by a coefficient of 10; see Equation (9).
FBC = (w_2_/w_0_)·10,(9)

Gelatin foaming capacity (FC) and foaming stability (FS) were determined according to Sathe et al. [88] with slight modifications. An amount of 1.0 g of the gelatin sample was weighed into a graduated cylinder and 50.0 mL of distilled H_2_O was added; the gelatin was dissolved in a water bath at 60.0 ± 1.0 °C while stirring. After dissolving, a dispersing instrument was placed below the level of the resulting solution and the solution was whipped at 10,000 rpm for 5 min. After whipping, the volume of the whipped solution was measured; FC (%) was calculated according to Equation (10). After standing at room temperature for 30 min, the volume of the whipped solution was measured again; FS (%) was calculated according to Equation (11)
FC = [(V_1_ − V_0_)/V_0_]·100,(10)
FS = [(V_2_ − V_0_)/V_0_]·100,(11)
where V_0_ is the original volume of liquid (50 mL), V_1_ is the volume of whipped solution (mL), and V_2_ is the volume of whipped solution after 30 min (mL).

Gelatin emulsifying capacity (EC) and emulsion stability (ES) were determined according to Neto et al. [89] with slight modifications. In a plastic test tube, 0.01 g of the gelatin sample was mixed with 5.0 mL of distilled H_2_O and after 10 s of thorough shaking, 5.0 mL of sunflower oil was added and shaken for 1 min at room temperature. The contents of the test tube were then centrifuged at 1000 rpm for 5 min. The height of the entire volume of liquid in the tube and the height of the emulsion were measured. The tube was placed in a preheated water bath at 55.0 ± 0.5 °C for 5 min, then the emulsion height was measured. EC (%) and ES (%) were calculated according to Equations (12) and (13)
EC = (h_1_/h_0_)·100,(12)
ES = (h_2_/h_0_)·100,(13)
where h_0_ is the height of the entire volume of liquid (mm), h_1_ is emulsion height after centrifugation (mm), and h_2_ is emulsion height after 5 min heating (mm).

The Moosavi–Nasab method [90] with modifications was used to determine the melting point; a solution of gelatin at the same concentration (6.67%) as after determination of gel strength and viscosity was used. A gelatin solution was introduced into a 3.0 mm diameter glass capillary to form a column at a height of 6.0 ± 1.0 mm. The sample capillary was allowed to cool at 10.0 ± 0.1 °C for 17 h (sol–gel transition). The capillary was then placed in a water bath at 10.0 ± 0.5 °C until it was completely immersed. The water bath was heated at 2 °C/min and the gelatin column in the capillary was monitored. The temperature at which the gelatin column began to move in the capillary (gel–sol transition) was recorded as the melting point (M_P_).

The Schrieber and Gareis method [26] with slight modifications was used to determine the gelation point; a solution of gelatin at the same concentration (6.67%) as after determination of gel strength and viscosity was used. The gelatin solution in the test tube was placed in a water bath. After warming to 30.0 ± 0.5 °C, ice water was added to the water bath until the cooling rate of the gelatin solution in the tube was 2 °C/min. Each time the temperature dropped by 0.5 °C, a metal ball weighing 0.10 g was inserted into the tube. The temperature at which the ball became stuck in or on the gelatin solution layer was recorded as the gelation point (G_P_).

## 5. Conclusions

*Cyprinus carpio* skeletons, formed as solid byproducts of carp filleting, are a valuable source of collagen. Through suitable processing using biotechnology to minimize the chemicals used, this byproduct represents a valuable and at the same time affordable source of renewable polymer suitable for the preparation of quality gelatins. These gelatins are comparable to other fish gelatins and can be used in the production of food, nutritional products, in pharmaceuticals (for example, macromolecular dosing matrices), and in the preparation of polymer films, coatings, gels, and cosmetics. By selecting suitable process conditions, the gelatin yields can be optimized. The processing technology of *Cyprinus carpio* skeletons responds to the challenges of the 21st century regarding the processing of polymeric byproducts into products usable in various applications. It respects the principles of circular economics, as the starting material can be used in high yield to prepare gelatins. The pigment isolated by centrifugation of the gelatin fractions can be used in the production of paints, fats for food production, and in cosmetic applications. Due to its high nitrogen content, the remaining undissolved residue after gelatin extraction can be used as an N-type fertilizer in agriculture. The maceration liquor remaining after demineralization can be recycled to calcium phosphate dihydrate, usable in feed mixtures. To further interpret the influence of the biotechnological collagen processing from *Cyprinus carpio* skeletons on the properties of gelatins extracted under different conditions in several extraction steps, it will be necessary to further study the chemical composition, amino acid content, molecular weight and secondary structure of different prepared gelatins.

## Data Availability

The data presented in this study are available upon request from the corresponding author.

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
