# Peer review of "Cyprinus carpio Skeleton Byproduct as a Source of Collagen for Gelatin Preparation"

_ijms, 2022, doi:10.3390/ijms23063164_

Round 1
Reviewer 1 Report
|
S.No |
Comment |
Response |
|
1. |
Title can be improved as “Cyprinus carpio skeletons by-product as a source of collagen- for gelatine preparation. |
|
|
2. |
Abstract grammar correction required L1 |
|
|
3. |
Grammatical check needed L 38. |
|
|
4. |
What is semi-unsaturated fatty acids? |
|
|
5. |
Line 116-118- lacks clarity. And the following sentence is not having continuity with the earlier statement. |
|
|
6. |
Line 123-What do you mean by starting tissue? |
|
|
7. |
Line 125-126- Does it mean alkaline or acidic condition, sentence is confusing. |
|
|
8. |
Discussions lack a proper justification, like what might be the reasons for low Gp of the gelatin prepared in the present study in comparison to other bovine or fish gelatins. |
|
|
9. |
Why the gelatins prepared from the 3rd and 4th fractions have good foam stability in comparison to the 1st and second fractions? |
|
|
10. |
The novelty of the work in adopting the different extraction steps for gelatine preparation is highly appreciated, but the gelatin fractions exhibited the different functional properties like viscosity and emulsion capacities differently. Whether it’s due to any structural disintegrity during the continous extraction process. Try to support with literature. |
|
|
11. |
During the multistage extraction process, whether subsequent temperature rise at each stage doesn’t influence the properties of the gelatine? |
|
|
12. |
Plates of the gelatine extracted at different stages can be shown. |
|
Author Response
Dear Sir / Madam,
Thank you very much for revision of our paper and detailed recommendations to improve it. We did our best to revise our paper according to your suggestions. Changes in the manuscript are highlighted in bold red color. Below, there are point-by-point responses to your comments.
Reviewer Comment No.1:
Title can be improved as “Cyprinus carpio skeletons by-product as a source of collagen- for gelatine preparation.
Response: We improved the title according to your recommendation: “Cyprinus carpio skeletons by-product as a source of collagen for gelatins preparation“. „Gelatins“ should be in plural as four fractions of gelatins are prepared. As the journal uses American English style “gelatins” without “e” is used (contrary to “gelatines” in British English).
Reviewer Comment No. 2:
Abstract grammar correction required L1
Response: Correction in the sentence was made.
Reviewer Comment No. 3:
Grammatical check needed L 38.
Response: Correction in the sentence was made.
Reviewer Comment No. 4:
What is semi-unsaturated fatty acids?
Response: Thank you for your comment. Our previously used term “semi-unsaturated fatty acids” is not correct. It was a result of inappropriate translation to English. We apologise for that. The correct term is “polyunsaturated fatty acids”; it is corrected in the manuscript (please see page 2 / line 81).
Reviewer Comment No. 5:
Line 116-118- lacks clarity. And the following sentence is not having continuity with the earlier statement.
Response: We revised both sentences to be clear and showing continuity; please see page 3 / lines 117-121.
Reviewer Comment No. 6:
Line 123-What do you mean by starting tissue?
Response: Starting tissue is the tissue (or organ) rich in collagen. We improved the beginning of the sentence; please see page 3 / line 123.
Reviewer Comment No. 7:
Line 125-126- Does it mean alkaline or acidic condition, sentence is confusing.
Response: Mostly, fish tissues (organs) are conditioning in acids. Alkaline treatment is used rarely. We have corrected the sentence to sound clear; please see page 3 / line 126.
Reviewer Comments Nos. 8, 9, 10, 11:
Discussions lack a proper justification, like what might be the reasons for low Gp of the gelatin prepared in the present study in comparison to other bovine or fish gelatins.
Why the gelatins prepared from the 3rd and 4th fractions have good foam stability in comparison to the 1st and second fractions?
The novelty of the work in adopting the different extraction steps for gelatine preparation is highly appreciated, but the gelatin fractions exhibited the different functional properties like viscosity and emulsion capacities differently. Whether it’s due to any structural disintegrity during the continous extraction process. Try to support with literature.
During the multistage extraction process, whether subsequent temperature rise at each stage doesn’t influence the properties of the gelatin?
Responses: All these comments refers to (results) and discussion part. The discussion and interpretation of the results was improved to cover all your questions and suggestions; 17 new references were added as well (the following references were re-numbered). For this purpose new chapter in Discussion was implemented, please see chapter “3.5 Discussion Summary“. Because of the complexity of discussion and interpretation of the results, we added in chapter “5. Conclusions” a final sentence outlining our future research which will be focused on detail study on chemical composition, molecular weight and secondary structures of prepared gelatins to get even better understanding of the influence of processing conditions on gelatins properties.
Reviewer Comment No. 12:
Plates of the gelatine extracted at different stages can be shown.
Response: For this purpose we put a photos of gelatin in the form of gelatin gel and gelatin sheet in Graphical Abstract. We had considered showing some other photographs of gelatin from some experiments, but finally we did not implemented them. To be honest, currently, we do not have 37 prepared samples of gelatins from our experiments to be photographed for this purpose; most of them had been used for the analysis.
Yours faithfully,
Pavel Mokrejš (corresponding author)
Reviewer 2 Report
The paper reports on the production of gelatin from Cyprinus carpio skeletons. The difference with previous studies relies on the nature of the fish (Cyprinus carpio). Skeleton by-products are used in line with the concept of circular economy. By reading the paper, I did not find any originality on the methods used to produce gelatin as well as for its characterization. It is known that the nature of the extracted collagen, and thus on gelatin, varies a lot from sample to sample. These effects depend on the change of the molar mass, the chemical composition (nature of the amino-acids) and the distribution of the amino-acids. A detailed investigation of these effects is very complicated due to the complexity of the involved chemical structures. It is thus known and expected that the properties of gelatin will vary depending on the origin of the product. Accordingly, the authors observed specific properties when analyzing the gelatin obtain from Cyprinus carpio skeletons by comparison with other studies based on other fishes.
The objectives of the study are clear. The experimental part is very well-detailed. The information is important for scientists who must select a source of gelatin with suitable properties to produce an appropriate material for a given application.
My understanding of the scope of the journal is that original results can be published regarding (1) fundamental aspects, (2) technical progresses and (3) application of novel technologies. Although this paper covers biology and chemistry, with perspective of applications in medicine, I don’t see any novelty for the 3 points just noted. My feeling is that the change of the nature of the fish is not sufficient to justify publication in this journal in the absence of any technological improvement or any improvement of the understanding of the processes. My fear is that researchers start to publish a new paper for any new fish or organ and that the journal becomes a place to publish datasheets (even though they can be important to communicate) rather than new scientific results. My recommendation is thus to publish this study in a more appropriate journal.
Minor Remarks
- The amount of HCL and enzymes (factors A et B) are written in %, which is not a unit for the concentration or the amount of the chemicals. Are they wt%?
- My first reading of Table 1 was not clear because the data are based on the experimental procedure, which is described more far away in the paper. It could be interesting to add a sentence to indicate to the reader where the reader can find the procedure shown in Scheme 2 (page 18)
Author Response
Dear Sir / Madam,
Thank you very much for comments to our manuscript. The paper was revised according to your remarks as well as according to Reviewer 1 recommendations. Completely new chapter “3.5 Discussion Summary“ was implemented to discuss and try to better interpret the results based on our findings with the support of literature; 17 new references were added in this context. Changes in the manuscript are highlighted bold red color. Below, there are responses to your comments.
Reviewer Comment No. 1: The paper reports on the production of gelatin from Cyprinus carpio skeletons. The difference with previous studies relies on the nature of the fish (Cyprinus carpio). Skeleton by-products are used in line with the concept of circular economy. By reading the paper, I did not find any originality on the methods used to produce gelatin as well as for its characterization. It is known that the nature of the extracted collagen, and thus on gelatin, varies a lot from sample to sample. These effects depend on the change of the molar mass, the chemical composition (nature of the amino-acids) and the distribution of the amino-acids. A detailed investigation of these effects is very complicated due to the complexity of the involved chemical structures. It is thus known and expected that the properties of gelatin will vary depending on the origin of the product. Accordingly, the authors observed specific properties when analyzing the gelatin obtain from Cyprinus carpio skeletons by comparison with other studies based on other fishes.
Response: As far as we are concerned the originality of our paper lies especially in enzyme conditioning of collagen tissue. This method is not only very rarely described in the literature but is not common in industrial practice of gelatin manufacturing process. The biotechnological procedure of gelatin extraction from poultry tissues was patented by our team: Patent CZ 307665 - Biotechnology-based production of food gelatine from poultry by-products (2019). Since 2019 the international patent application of the invention under the same name (PCT/CZ2018/050054) is the subject of a research assessment. We adopted and modified the biotechnological process to fish tissue. Further, the procedure is proposed to minimise the residual unprocessed substance after extraction of four gelatin farctions as much as possible; for this purpose multiple extraction process was applied. In addition, there are other product arising during the processing which can be further utilised. All of this moves our procedure close to the „zero-waste“ technology. A slight improvement of chapter “5. Conclusions” was made to outline future research in this topic.
Reviewer Comment No. 2: The objectives of the study are clear. The experimental part is very well-detailed. The information is important for scientists who must select a source of gelatin with suitable properties to produce an appropriate material for a given application.
Response: Thank you very much for the positive evaluation of our experimental part.
Reviewer Comment No. 3: My understanding of the scope of the journal is that original results can be published regarding (1) fundamental aspects, (2) technical progresses and (3) application of novel technologies. Although this paper covers biology and chemistry, with perspective of applications in medicine, I don’t see any novelty for the 3 points just noted. My feeling is that the change of the nature of the fish is not sufficient to justify publication in this journal in the absence of any technological improvement or any improvement of the understanding of the processes. My fear is that researchers start to publish a new paper for any new fish or organ and that the journal becomes a place to publish datasheets (even though they can be important to communicate) rather than new scientific results. My recommendation is thus to publish this study in a more appropriate journal.
Response: To be honest, I agree with you that there are hundreds papers describing processing of different kind of fish and their organs into collagenous products (gelatins, hydrolysates, peptides). That is why we tried to be different from these studies. To the above mentioned explanation of the novelty of our work another argument can be added. Many studies focused on the influence of technological condition on gelatin yield and basic properties of gelatin (gel strength, viscosity, ash content), some studies discuss the results of molecular weight; in other studies amino acid composition or structures of gels (using e.g. SEM) are analysed. Detailed analysis covering gel-forming and surface properties of gelatins is missing. These are very important for practical applications of gelatins especially in food and pharmacy. The aim of our paper was to cover as much as of these gelatin properties to give detail information for the practice and scientists as well. Finally, I got the invitation from Section Managing Editor of International Journal of Molecular Sciences to publish featured article in special issue "Smart Biobased Polymers and Composites: Current Challenges and Opportunities" and I promised to offer this kind of manuscript.
Reviewer Comment No. 4: The amount of HCL and enzymes (factors A et B) are written in %, which is not a unit for the concentration or the amount of the chemicals. Are they wt%?
Response: Yes, the concentration of HCl and the amount of enzyme is in wt%. It was corrected in the text. Concentration of chemicals is expressed in mol/l, but as well as in weight per-cent. As for acid we decided to use wt% as this unit is common to express concentration of HCl during demineralization in industrial production of gelatins from mammalian bones; the other authors use wt% as concentration unit of acids during demineralization procedure. We decided to express the amount of enzyme in weight per-cent as it is practical for dosing enzyme on the weight of processed collagen.
Reviewer Comment No. 5: My first reading of Table 1 was not clear because the data are based on the experimental procedure, which is described more far away in the paper. It could be interesting to add a sentence to indicate to the reader where the reader can find the procedure shown in Scheme 2 (page 18).
Response: We agree with you. The unclear feeling may be due to the fact that (according to journal template) the “4. Materials and Methods” chapter is not prior to “2. Results” and “3. Discussion” chapters. We improved the clarity for the potential reader according to your suggestion. At the beginning “2. Results” chapter there is a new sentence referring to procedure of processing of Cyprinus carpio skeletons into four gelatins fractions presented in Scheme 2 in chapter “4. Materials and Methods”.
Yours faithfully,
Pavel Mokrejš (corresponding author)
Round 2
Reviewer 2 Report
I thank the authors for their answers to my concerns and for the update of their paper, and especially the addition of a paragraph 3.5 with the summary discussion.
After reading the first version, I considered that the technical quality of the paper was high and that the paper mentioned new data, which should be important for researchers working in the field. Beyond novelty, I had concerns on scientific originality, which is something quite different.
The addition of the new paragraph helps the reader, which is not a specialist in the field, to better understand the complexity of the involved chemistry, the scientific issues and the differences between the other works previously published. This paragraph gives a more scientific and less technical understanding of the experimental results.
All my other remarks were considered. I understand that the authors applied the template of the journal.
I have very minor new remarks
It is recommended to apply the IUPAC nomenclature rules. Please use molar masses rather than the old version (molecular weights)
Line 674 = medium and not medium
My conclusion is that the answers of the authors are convincing and that I can join the other reviewer by recommending publishing the paper.